# An Exploratory Study on the Conceptualization of Burnout among the Professional Esports Athletes: Focused on League of Legends Champions Korea League

**DOI:** 10.3390/healthcare12111127

**Published:** 2024-05-31

**Authors:** Hyoyeon Ahn, Inwoo Kim

**Affiliations:** 1Department of Physical Education, College of Education, Seoul National University, Seoul 08826, Republic of Korea; ahy0522@snu.ac.kr; 2Department of Sports Culture, College of the Arts, Dongguk University, Seoul 04620, Republic of Korea

**Keywords:** professional esports athletes, burnout, exploratory research, conceptualization

## Abstract

Research on the stress and burnout experienced by athletes in the esports field remains limited, necessitating an approach that considers the unique environment and circumstances of esports athletes. This study aims to explore the conceptualization of burnout experiences among professional esports athletes. The participants included 88 athletes from the League of Legends Championship Korea (LCK). Among these, in-depth interviews were conducted with 7 athletes who had experience in international tournaments (the World Championship), while an open-ended survey was completed by 81 athletes. Data collected through in-depth interviews and an open-ended survey were digitized and utilized for analysis. Through critical review by another author and inductive categorization, the conceptual components of esports athletes’ burnout were derived. Through the analysis of 251 raw datum, it was conceptualized into five conceptual factors: performance, overtraining, interpersonal relationships, physical and psychological exhaustion, and career and motivation. The results of this study confirm that esports athletes, like athletes in other conventional sports, experience burnout during their careers, highlighting issues in their unique environment, particularly in aspects of interpersonal relationships and training and rest conditions. This research can serve as a foundational resource for effective athletes’ psychological health management in the esports field and underscores the need for further research on burnout among esports athletes.

## 1. Introduction

In the past, esports made its debut as a demonstration event at the 2018 Palembang Asian Games and gained significant attention by being officially adopted as a medal event at the 2022 Hangzhou Asian Games. Moreover. There have been plans discussed in the media to promote e-sports to an Olympic event, recognizing its establishment as an integral aspect of modern leisure activities. The International Olympic Committee (IOC) has outlined its commitment to fostering the growth of esports in the “2020 + 5 Agenda” [1]. Notably, starting with the 2023 Singapore event, the IOC has transitioned from the previous “Olympic Virtual Series (OVS)” nomenclature to “Olympic Esports Series (OES)”, signaling a clear acceptance of esports within the IOC framework.

In the Olympic Esports Series (OES), video games simulating real sports or virtual reality (VR) competitions of actual sports have been incorporated. This utilization is considered effective in overcoming constraints such as venue availability, transportation, and facility limitations, making VR-based esports an anticipated recreational activity in the leisure pursuits of contemporary individuals. Notably, esports titles like League of Legends (LoL) have garnered global attention, reaching a level comparable to conventional sports leagues such as the NFL and NBA. LoL is a MOBA (Multiplayer Online Battle Arena) genre in which two teams aim to occupy the opposing area in a 5-on-5 team match. The term “esports” has undergone sportification [2], aligning with specific video games. Hence, referring to participants in specific esports disciplines as “athletes” provides a comprehensible framework for understanding and examination.

In Korea, the LoL Champions Korea (LCK) conducts two leagues annually (Spring and Summer leagues), following a global schedule. Similar to conventional sports like baseball, soccer, and basketball, esports in Korea has evolved into a franchise system with major corporate investments, operating ten teams. Each team operates up to the second division, participates in the second league, and some even engage in a third-division Academy League for talent scouting. Esports athletes also experience stress and anxiety similar to conventional athletes [3], but because they have characteristics that distinguish them from conventional sports, such as a sedentary lifestyle, an exploratory research approach is necessary.

Like conventional sports, esports athletes undergo team dynamics influenced by victories and defeats, as well as psychological and behavioral changes due to competition-related stress. During the entire season, communal living in the camp is common, presenting psychological and physical challenges to interpersonal relationships among teammates and coaches. In the case of LCK, the regular league (Spring and Summer leagues each) consists of ten teams that operate in two rounds of the full league, with a best-of-three format. Typically, one game lasts 30 to 40 min, and on non-game days, six unofficial scrimmage matches are usually held. Because of these conditions, esports athletes are exposed to competitive situations many times during the entire season. In addition, given the absence of real-time coaching or feedback during matches, players emphasize self-management of psychological well-being during competitions. Consequently, esports is being introduced as a category of sports where psychological management is important, emphasizing the necessity for a sports psychology approach [4]. The exploration of burnout conceptualization among professional esports athletes has not been conducted yet. Thus, it is worth examining as foundational information for the psychological health management of esports athletes, who possess unique characteristics compared to conventional sports athletes.

The growing interest in burnout among coaches, not just athletes, in Korea has led to studies examining the differences in stress types and burnout experiences based on specific sports disciplines [5,6,7,8]. The exploration of burnout concepts, rooted in the well-established Maslach’s burnout framework [9,10] from general psychology, is being adapted for specific professions and environments through prior research [11,12]. Raedeke and Smith have established the concept of burnout among athletes and coaches in sports field [13], and interest continues for every sports event such as taekwondo and baseball [14,15]. The importance of managing athletes’ stress is explained in McGrath’s four-stage stress model [16], which illustrates that stress can become a vicious cycle if not properly coped with. Additionally, Smith [17] has described how the conceptual model of burnout parallels the stress model, explaining the relationship between stress and burnout in athletes. The concept of recovery from fatigue in athletes is categorized into physical and psychological aspects [18]. It is expected that the psychological fatigue of e-sports athletes will be relatively higher than the physical fatigue of conventional athletes. In fact, because performance in esports is mainly expressed through motor control skills such as mouse and keyboard operation, a different type of fatigue will occur than in conventional sports. Accordingly, it is deemed necessary to examine the concepts of burnout in the context of the environment and situations specific to e-sports athletes.

The perception of esports remains contentious, especially in light of the World Health Organization’s inclusion of “gaming disorder” in the International Classification of Diseases (ICD)-11 [19]. Despite the ongoing debate, there is a lack of reported psychopathological issues among esports athletes. Recent studies have explored the benefits of video game participation, highlighting the need for bidirectional studies on positive and negative aspects in follow-up studies [20,21,22]. The psychological difficulties that athletes experience on the actual sports field are explained by the conceptual theory of sports competition anxiety [23]. Generally, since not all athletes’ experiences of stress are classified as mental illnesses, the term “gaming disorder” still lacks sufficient evidence.

Therefore, it is necessary to examine cases based on the exploration of the experiences of esports athletes in unique environments, like in this study. Particularly, understanding their difficulties serves as a basis for proper application in future athlete management. Hence, the purpose of this study is to examine cases of difficulties faced by esports athletes in the field and explore the concept of burnout among e-sports athletes. This approach has meaningful foundational data for managing esports athletes, who are gaining attention as a form of sports content today. Ultimately, it is believed that this can contribute to the quality of life and psychological health of esports athletes.

## 2. Methods

### 2.1. Participants

To explore what e-sports professional athletes experience and perceive as burnout due to stress, in-depth interviews were conducted with 7 current or retired LCK athletes who have over 5 years of playing experience and have represented their country in international competitions (current, 3, and retired, 4). All the athletes who participated in the in-depth interviews have a history of competing in the LoL World Championship, and their characteristics, such as playing career and highest achievements, are presented in Table 1 below.

In addition, an open-ended survey based on the in-depth interviews was conducted among 81 athletes (current, 73, and retired, 8) with at least 2 years of LCK experience (in-depth interview participants are not included). All research participants were male players registered with the Korea e-Sports Association, either current or retired professional athletes. The average age of all participants was 23.4 years, and the average playing career was 2.4 years.

### 2.2. Procedure and Data Collection

This study was conducted with ethical approval (IRB No. 2202/002-007), and all research processes were carried out with prior consent obtained. The procedure of the study involved conducting a “literature review’ and “in-depth interviews”, followed by an “open-ended survey”. The collected data were interpreted through inductive categorization to “establish conceptual constructs”. Research participants were recruited by delivering the research explanation through the team coaches or front desk staff, and data collection was conducted for 1 year and 8 months due to difficulties in recruiting research participants. The interviews were conducted separately by the authors and recorded using a smartphone application. The open-ended survey was conducted by the author, who visited the team that had been cooperating in advance, and the athletes who agreed to participate were surveyed.

Initially, in the literature review process, data on burnout and stress among professional esports athletes were collected. Based on this review, semi-structured, in-depth interview questions were developed through researcher meetings with field practitioners and experts (2 coaches and 2 sports psychology doctors). For example, asking questions such as “What is the most difficult thing about being an esports athlete?” and “What are the causes of stress or burnout experiences?”. Subsequently, 1:1 in-depth interviews were conducted with current or retired professional athletes who had international competition experience (LoL World Championship), each lasting about 60 min. The recorded data from these interviews about the athletes’ burnout experiences were transcribed and used in expert meetings. Based on in-depth interview data, we conducted an open-ended survey using more specific and detailed questions. Specifically, the questions were designed to allow responses about the causes of burnout experienced during their athletic career and the physical and psychological symptoms they experienced when they became aware of the burnout. In details, for example, “Q. When did you most often experience stress during your career as an athlete?”, “Q. What is most memorable experience of stress or burnout? The reason is that?”, and “Q. What symptoms or behaviors did you exhibit when you experienced severe stress and burnout during your athlete career?’” The survey was conducted with the cooperation of several LCK teams and by contacting retired athletes individually. The collected open-ended survey data were coded digitally and used by the researchers for inductive categorization during meetings.

### 2.3. Data Analysis

In this study, in-depth interviews and an open-ended survey were conducted to understand the burnout and stress experienced by professional esports athletes. Through in-depth interviews and open-ended surveys, we categorized the common burnout experiences of esports athletes and understood their burnout conceptual structure.

The collected data were analyzed qualitatively, referencing the six-step procedure of Creswell [24]. First, the in-depth interviews were transcribed into documents, and key concepts were derived from the transcription data. During the transcription and meaning extraction process, the raw data were transcribed as is, including any errors in vocabulary or grammar, to avoid distortion. The data collected through open-ended surveys were coded, and inductive categorization was conducted through researcher meetings. The qualitative data analysis was performed with a member check for mutual critical review among researchers to ensure the authenticity of interpretation [25]. Finally, the derived conceptual structure was reviewed for the validity of data analysis and interpretation through a researcher meeting, and the final goal is to understand the common categories of the collected data and the conceptual structure of e-sports athletes’ burnout.

## 3. Results

### 3.1. In-Depth Interview

Through in-depth interviews with seven professional esports athletes, we looked in depth at the causes and symptoms of burnout. It was helpful for understanding the stress situations and burnout experiences of professional e-sports athletes and was used as basic data in the creation of open-ended survey questions and the inductive categorization process. The most frequently mentioned common experiences related to the causes and symptoms of burnout in the interviews were “performance”, “overtraining”, and “interpersonal relationship”.

#### 3.1.1. Performance

Through in-depth interviews, it was confirmed that esports athletes experience the same difficulties as conventional sports in terms of their performance and competition for starting positions. Although players long for a call-up to the first division, some responded that the pressure of being sent down to the second division also affects their playing style. And there was an answer to the psychological difficulties experienced when being pushed out of the starting player competition.

“*After send-down and call-up system were created, there was a climate of competition for the starting player and position, but I felt a bit burdened by that. Of course, I work hard, but I think that if my performance is not good, another player may play next match, so I try to play safely in the match. (omitted) I need to focus more on trying to play the game well, but I think I’m choosing to play it safe to avoid making mistakes or failing.*”<Athlete E>

“*I thought I was good at the game, but since I haven’t had a chance to play since the last game, I think the gap between me, and OO is growing. It would be nice to be given the opportunity to play as a starting player, but I’m not having fun as a player these days. Because I don’t even play in scrimmages and only plays rank games alone... (omitted) but LoL is a team game, so I feel like I’m falling behind more and more, and I barely talk to other people, so vigor has disappeared.*”<Athlete F>

#### 3.1.2. Overtraining

Through in-depth interviews, the difficulties caused by overtraining of players were confirmed. Due to the characteristic of esports, it is necessary to pay attention to the training schedule that lasts from afternoon to dawn. In fact, there were responses about difficulties due to the characteristics of esports, which did not take into account the training schedule and recovery time that are different from those of conventional sports.

“*I’ve been a professional player for a long time, and I don’t think I’ve ever been forced to train at late night like this season. (omitted) If I am forced to do personal training at late night, my efficiency drops. The current team atmosphere is that the coach just tells them to sit in front of their PC for a long time, so it seems like the other players are forced to just sit there without any motivation. (omitted) but the head coach created a coercive climate system where all players had to practice until 4 a.m., so it was difficult.*”<Athlete A>

“*I usually relieve stress at home during my vacation, but recently head coach cancel our weekend vacation and order to practice, so I’m having a hard time. (omitted) However, since the coach was not satisfied with the team’s performance in the last match, there will be no vacation this month. Because I live in a camp without any vacations (involve sleepovers), the frustrating sometimes make me more difficult. I feel like I can’t sleep comfortably, and that’s why I can’t manage my stress.*”<Athlete C>

#### 3.1.3. Interpersonal Relationship 

Through in-depth interviews, difficulties arising from the players’ interpersonal relationships were revealed. Due to the characteristics of training camps and group living throughout the year, problems due to interpersonal relationships are expected to be frequent, and issues of peer relationships and coach–athlete relationships are already being studied in conventional sports.

“*In every feedback meeting, player OO seems to blame other players’ mistakes as the cause of team’s defeat rather than admitting his own mistakes. This is because in the case of young players, the feedback meeting ends without even being able to express their dissatisfaction, which can lead to mental illness later on. (omitted) That player is younger than me, but I feel like he’s being rude to me, so sometimes I can’t concentrate in the game because I’m worried about him. I treated him well first, but I think he’s acting carelessly.*”<Athlete B>

“*I think the head coach doesn’t like me. He always says that he will select me as a starting player if I raise my personal rank (on the game), but he doesn’t seem to check my personal rank records. He doesn’t even seem to make eye contact with me, and since I haven’t even played in scrimmage matches lately, I really hate going to the arena (stadium) as a substitute player. I’m really confident that I’ll do well, and I’ve never expressed dissatisfaction with the coaching staff’s guidance, but I feel like they’re being too careless with me.*”<Athlete D>

### 3.2. Open-Ended Survey

The data collected on the causes of burnout through in-depth interviews and open-ended surveys were organized into five conceptual factors through prior research and expert meetings. Specifically, as a result of inductive conceptualization of the 251 raw datum collected through open-ended surveys, it was confirmed that the factors ranked in the order of “performance (77)”, “overtraining (60)”, “interpersonal relationship (32)”, “physical and psychological exhaustion (46)”, and “career and motivation (27)”. In addition, cases with less than five responses were classified as “other (8)” and excluded from the construct concept in this study. In the process of analyzing the data collected through an open-ended survey, the factors “physical and psychological exhaustion” and “career and motivation” were added. This point is explained by the fact that open-ended questionnaires are more universal than the in-depth interviews conducted previously.

#### 3.2.1. Performance

As a result of extracting factors through inductive categorization of the 251 raw datum, the performance factor appeared most frequently. The sub-concept contents regarding performance were categorized into “team performance (21)”, “individual performance (42)”, and “call-up and send down (14)” through expert meetings. In fact, it was confirmed that the decline in individual performance was a primary factor in burnout among esports athletes, and it showed the highest frequency among other sub-concepts. Additionally, since LoL is a team sport, it was observed that players are influenced by the performance of peer athletes and the team. There were also responses related to competition for starting positions due to the league’s operation of the call-up and send-down system.

#### 3.2.2. Overtraining

The main contents of overtraining were categorized into three aspects: “overtraining schedules (34)”, “coercive team climate (12)”, and “lack of rest (14)”. In many responses, athletes expressed concerns about the burden of training volume and schedule, including compulsory individual practice, streaming, and tight schedules for scrimmage matches. In fact, the frequency of primary data related to excessive training schedules was the highest.

#### 3.2.3. Interpersonal Relationship

The main contents of the interpersonal relationships factor were categorized based on the target of the relationship, with responses pertaining to “coach–athlete relationships (19)” and “peer relationships (13)”. The responses mainly concerned conflicts arising during team feedback meetings and communal living in training camps, and it was observed that troubles with coaches were more frequent than those with peer athletes. Since athletes primarily live together in accommodations during the season, the type of interpersonal relationships appeared limited to these two categories (Table 2).

#### 3.2.4. Physical and Psychological Exhaustion

Responses indicating persistent fatigue were high, and through expert meetings, these were categorized into “physical exhaustion (20)” and “psychological exhaustion (26)”. The responses about fatigue were organized under the concept of “exhaustion”, a factor of burnout according to Maslach’s burnout theoretical concept, with both physical and psychological exhaustion appearing at similar frequency. Psychological exhaustion included stress due to criticism from fans, negative public opinion on social media, and ongoing burnout caused by failure in stress management.

#### 3.2.5. Career and Motivation

The main responses in the area of career and motivation factors included issues related to military service and retirement concerns, categorized as “career (16)”, and aspects of personal will and loss of interest in gaming, categorized as “motivation management (11)”. Particularly, the process of applying for a deferment of military service duty during the season, including preparing necessary documents, was reported to negatively impact players’ career aspirations and level of motivation. This conceptual category predominantly reflected a high frequency of responses influenced by factors external to the game, such as athlete contract issues and military service duty.

#### 3.2.6. Others

There were also responses categorized as “others”, which were not frequent enough to be classified as common symptoms or causes. These included issues related to individual health problems, such as back diseases. Through discussions between researchers, pain or difficulties caused by disease can lead to psychological stress or exhaustion, but it was classified as not a common case. Additionally, some athletes reported experiencing obsessive behavior regarding setting up their devices (mouse, chair, etc.), and there were cases where dissatisfaction with salary or contract options was cited as a cause of burnout. These cases were also categorized as “others” as they were not common.

## 4. Discussion

Based on the data collected through in-depth interviews and open-ended surveys, we derived the conceptual framework of burnout factors among esports athletes. Ultimately, these were organized into the following five conceptual factors: “performance (30.7%)”, “overtraining (23.9%)”, “interpersonal relationships (12.7%)”, “physical and psychological fatigue (18.3%)”, and “career and motivation (10.7%)”. The interpretation of the results from the data analysis and the ensuing discussion are as follows:

The first conceptual factor in esports athletes’ burnout is “performance”. As the data were collected from professional athletes, the raw data related to athletic performance were found to have a high frequency (30.7%) of actual causes of stress and burnout. Through these results, it was confirmed that esports athletes, like other athletes in regular sports, experience competitive anxiety due to wins and losses and respond sensitively to management and changes in their performance. Through this, the validation of the application of psychological skills training based on sports psychology theory to esports athletes was supported, and it was confirmed that performance is a major area in the management of athletes’ psychological health and well-being.

The second conceptual factor is “overtraining”. Due to the character of the esports field, all training, including scrimmage matches and personal practice, takes place in front of one’s PC. Accordingly, like athletes in general sports, they are not given time to ventilate and rest in between moving to the stadium and changing locations for each training session. Compared to conventional sports, the importance of recovery may be overlooked due to the character of esports, which are limited to a sedentary lifestyle in front of a PC. Usually, individuals do it on their own, but it was mostly limited to one’s life at their PC. In this environment, raw data such as “frustration” and “lack of free time” appear to have been collected, as in the derivation of this conceptual factor. In addition, there were responses about “training amount”, and data such as “excessive training time” and “lack of rest” were collected. Although there may be slight differences depending on the philosophy of the coaching staff of each team, it is common for teams to train until late at night when online server usage is optimal and practice games with professional athletes from other teams are possible. Unlike the concepts of burnout developed in conventional sports contexts [5,13,14,15], overtraining has been categorized as a distinct conceptual factor. Previous studies are already being conducted to explore burnout predictors among athletes based on the characteristics of a specific sport event, such as swimming [26,27], soccer [28], and rugby [29,30]. This unique environment and training pattern suggest the need for research into customized management strategies that consider esports athletes’ health and biorhythm.

The third conceptual factor is “interpersonal relationship”. In the field of sports psychology, team cohesion and communication, including coach–athlete and peer relationships, are also major research topics [31,32,33]. This study also identified burnout factors (stressors) related to interpersonal relationships experienced by esports athletes. The findings, such as “conflicts among players” and “relationships with coaches”, are not different from those in conventional sports [13,34]. However, the data collected showed that conflicts, disagreements, and hurt feelings mainly occurred during “team feedback meetings”, which can be seen as a unique aspect of the esports training environment. In fact, e-sports teams often have unofficial scrimmage matches scheduled with other teams on non-game days, separate from the number of league schedules. Typically, six scrimmage matches are played per day, and if deemed necessary by the coach, additional games may be conducted at mid night or dawn. In different countries, depending on the number of league teams, the format varies. For instance, each LCK team plays a minimum of 72 to a maximum of 108 matches per year. Including playoffs and scrimmage matches, esports athletes are exposed to a high number of competitive situations. This suggests that psychological exhaustion can occur during the team feedback meetings typically held after each match. While other sports might resolve issues in one or two feedback meetings per day, the frequency of these team meetings in esports, due to the number of matches, can create an environment prone to conflicts during non-game situations.

In addition, esports differs from conventional sports in that talented youth athletes are often scouted directly into the second or third teams of the professional league without going through the school elite or club sports system. As a result, players are exposed to communal living and competitive activities at a young age. This characteristic of esports leads to a culture where athletes are typically valued based on their performance only, regardless of age. This can result in a lack of courtesy or consideration among players. While the climate and personality may vary from team to team or individual to individual, it is believed that issues in interpersonal relationships in esports require a different perspective from those in conventional sports. Thus, the efficiency of communication between athletes and with coaches becomes crucial, and interpersonal skills emerge as an important life skill for these athletes.

The fourth conceptual factor is “physical and psychological exhaustion”. As previously mentioned, esports athletes generally have a lot of matches compared to other sports. Additionally, esports have irregular game times depending on each team’s game strategy. And matches, which usually last about 50 min, are played at least six times a day for scrimmage matches and at least three times a day for the actual league schedule. Considering the time spent preparing for matches, athletes are subjected to relatively long periods of tension and competitive anxiety. This study collected the raw data related to physical symptoms such as back and wrist pain. Since esports athletes participate while sitting, their physical activity and calorie expenditure may be low, but prolonged concentration and sitting can lead to a different kind of fatigue. Indeed, this prolonged sedentary lifestyle is associated with spinal disorders, and most athletes show high interest in spinal care and maintaining correct posture.

Psychological fatigue was also a notable finding in the collected raw data from the interviews and survey, reflecting characteristics distinct from conventional sports. As previously mentioned, esports athletes, due to the nature of their sport, spend most of their day in the practice room, or in front of their PC. And since LoL is a five-player team sport and everyone lives according to the common training schedule, the daily schedule can be accepted passively or forcefully. This can result in feelings of confinement. Additionally, there were responses from “online communities” and “social network”, and since esports athletes spend most of their time in front of their PC, they may be more sensitive to online responses and discussions about games and one’s performance. Unlike conventional sports events that require a rest period for physical fatigue after a game, there is no prior research on the need for rest after a game in esports events. In particular, because it is necessary to distinguish between physical and psychological fatigue in athletes [18], it is necessary to focus on the relatively high psychological fatigue of esports athletes. This suggests that the psychological health management of esports athletes should be a consideration.

The next conceptual factor is the “career and motivation” issue. Due to the unique circumstances of the Republic of Korea, all Korean citizens are required to serve in the military. Accordingly, military duty issues are one of the concerns of all athletes in their 20s, and through this study, it was found to be a major burnout factor for esports athletes. In particular, unlike other sports (in the case of other sports, there are relatively more military duty exemption benefits for excellent athletes), there is no provision for esports athletes to continue their careers during military duty, often leading to career termination (retirement) rather than a temporary interruption. This issue impacts athletes’ career aspirations and motivation levels, and as the time to serve during their prime approaches, some young players contemplate retirement regardless of their wishes. While conventional sports may deal with issues related to aging curves and physical performance, esports athletes often face the unique situation of having to pause their careers when they are still physically fit and skilled. In Korea, the six athletes who won gold medals at the Hangzhou Asian Games were granted exemption from military duty, but this applies only to a few players. In fact, there have been reports that Korean athletes tend to prioritize military exemption [35], and it has been confirmed that esports athletes are no exception. Therefore, coaches need to consider the issue of military duty when managing the mindset of their athletes.

In addition to the five conceptual factors of esports athletes’ burnout previously mentioned, some responses indicated that issues such as “salary” and “setting compulsion” experienced while living as a player could be causes of burnout. However, these were considered minority opinions in the process of extracting raw data and were not categorized as sub-conceptual factors. Nonetheless, it was confirmed that e-sports athletes do experience stress related to “salary and contracts”. Additionally, unlike the pre-performance routines or superstitions of baseball players or golfers, esports athletes invest time in the “devices setting” of their chair, mouse, and monitor. Even though these data points represent a minority opinion, they are considered valuable for further research to explore the psychological well-being of esports athletes.

## 5. Conclusions

The purpose of this study was to explore the burnout experiences of esports athletes and exploring the conceptual framework of esports athlete burnout through an in-depth interview and an open-ended survey. This enhanced the understanding of esports and confirmed that esports athletes experience similar sports competition anxiety as athletes in conventional sports, facing stress due to performance and overtraining. Therefore, this study suggests the applicability of sports psychology research to esports athletes and provides direction for research aimed at promoting their psychological health and well-being.

Firstly, the qualitative analysis of in-depth interview data revealed that burnout could be induced by issues related to personal or team performance, overtraining, coach-player and peer relationships, physical and psychological exhaustion, and problems related to career and motivation, which are similar to conventional sports.

Secondly, based on in-depth interviews, an open-ended survey was conducted, establishing five conceptual factors: performance, overtraining, interpersonal relationships, physical and psychological exhaustion, and career and motivation. Among these, performance and overtraining were identified as having high frequencies.

Thirdly, the analysis of data collected through in-depth interviews and open-ended surveys showed that while the content of the burnout conceptual framework for esports athletes is not significantly different, it possesses unique characteristics compared to conventional sports. This highlights the need and direction for further research.

Fourthly, our study has several limitations, and these must be taken into consideration in order to apply and expand the research results. This study was conducted specifically in the League of Legends (LoL) esports discipline. While there are no esports of similar international franchise and league operation scale with annual international competitions, the study is limited by the characteristics of LoL (a team sport with specific schedules). Whether the burnout conceptual framework from this study applies to all esports disciplines should be addressed in future research. In addition, because the number of athletes is not large compared to conventional sports, retired athletes were included as research participants in order to sample and explain the specific characteristics of esports. And the fact that the data collection period for this study was conducted over a long period of more than a year must also be taken into consideration in the interpretation. Therefore, these aspects need to be taken into consideration in follow-up research. Lastly, this study, being qualitative, only explored the conceptual framework of burnout factors among esports athletes. Subsequent quantitative research based on this study, aiming to measure these concepts, could provide more practical assistance in the field.

## Figures and Tables

**Table 1 healthcare-12-01127-t001:** Characteristics of the in-depth interview participants.

Type	Participants	Career (yrs.)	WC Participation (Times/Highest Achievement)
In-depth interview	A	7	3/2nd
	B	8	4/2nd
	C	5	1/quarterfinal
	D	5	1/quarterfinal
	E	7	3/1st
	F	5	1/quarterfinal
	G	7	4/1st

Note: WC refers to LoL World Championship; “times” refers to participation number of WC.

**Table 2 healthcare-12-01127-t002:** The conceptual factors of esports athletes’ burnout.

Details	Sub-Concepts	Concepts (*n*)	Percent
Consistent lower league ranking	Team performance (21)	Performance (77)	30.7%
Continued losing streak			
Performance continues to drop	Individual performance (42)		
Painful of my faults in the match			
Anxiety about whether I can do well			
Concern about send-down	Call-up and send-down (14)		
No consideration for call-up			
Compulsory personal practice	Overtraining schedule (34)	Overtraining (60)	23.9%
Late-night schedule and training			
Sustaining a coercive team climate	Coercive team climate (12)		
Daily life without freedom			
Not freely even during rest	Lack of rest (14)		
When there is no holiday			
Discord with coaches	Coach–athlete relationship (19)	Interpersonal Relationship (32)	12.7%
Different opinion from the coaches			
Feel like the coaches hate me			
Individualistic playing of peer athlete	Peer relationship (13)		
Conflicts of opinion with others			
Negative attitude towards me			
Fatigue persists and does not recover	Physical exhaustion (20)	Physical and psychological Exhaustion (46)	18.3%
Severe back or wrist pain			
Lack of sleep due to sleep disorder			
Continued failure to manage stress	Psychological exhaustion (26)		
Feel like I’ve been tired for a long time			
Criticism from fans on social media			
Postponing military service	Career (16)	Career and Motivation (27)	10.7%
Worry about retirement			
More athletes younger than me			
Contracts become more difficult	Motivation regulation (11)		
The game is no longer fun			
Lumber pain recurrence		Others (8)	3.1%
Large salary gap compared to others			
Hard time due to obsession with setting			

## Data Availability

The data are available upon request from the corresponding author.

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
