# Peer review of "An Exploratory Study on the Conceptualization of Burnout among the Professional Esports Athletes: Focused on League of Legends Champions Korea League"

_healthcare, 2024, doi:10.3390/healthcare12111127_

Round 1

Reviewer 1 Report

Comments and Suggestions for Authors

After carefully reading the manuscript I received for review, I found the subject of the research presented in it to be very interesting. In recent years, there has been an increasing interest in the results of scientific research on e-sports athletes.

Based on data collected through in-depth interviews and an open-ended survey, the authors derived a conceptual framework of burnout factors among e-sports athletes. Ultimately, they were divided into five conceptual factors: performance, overtraining, interpersonal relationships, physical and mental fatigue, and career and motivation. In my opinion, the interpretation of the results would be more complete if the authors supplemented the data analysis by showing the percentage share of individual subfactors and factors in the answers given by the 81 surveyed e-sports athletes. Such an addition may be adding percentages to the content in Table 2 or creating a list presenting the number of responses and percentages in the form of a chart. Adding quantitative analysis to the qualitative analysis of the obtained results will allow readers to better understand the presented interpretations and ultimately determine the conceptual framework of burnout factors in the study group.

My second comment concerns the suggestion to change the title to consider the specificity of the respondents’ environment. I suggest changing the title to: An Exploratory study on the Conceptualization of Burnout among the Korean Professional Esports Athletes. This is due to the problems and concerns related to military service obligations, which are widely described in the discussion, and which do not apply to e-sports athletes in other countries. The specific social demands placed on men living in Korea may influence the importance of the 'Career and Motivation' factor in burnout among e-sports athletes.

Reviewer 2 Report

Comments and Suggestions for Authors

An Exploratory study on the Conceptualization of Burnout among the Professional Esports Athletes

 Dear Authors

I enjoyed reading your manuscript. Please find my comments below.

ABSTRACT

How were the views of the experts collated and compared? Who were the experts.

INTRODUCTION

L53-55: : provide a reference to support the statement.

L57-68: This paragraph provides some useful context for your results, but more is needed. Please provide more information on the participants training environment (before the results section) and scheduling. For example, in a typical week what is the volume and frequency of games per week for how many weeks?  Provide some background on competition incentives for both players and coaches that may relate to expectations, outcomes and career development. How does this relate to team progression in the league and or points structure (or threats to career development).

L86-89: This sentence needs rewording – possibly split into 2 sentences.

L69-78: Would be useful to provide a brief description of the Burnout concepts here and how they might relate to e sports participants.

L76-79: This sentence needs more detailing in providing a rationale for why esports athletes demands and / or population type may lead to specific burnout symptoms, different from gross motor control sports.

The aim as stated (in the final paragraph of the introduction) seems partly both inductive and deductive. It cannot be both. This is an inductive study, as indicated in the method.

METHOD

2.1 Participants

How were – for both interviews and survey – the participants recruited? Were the recruitment methods the same or different for both?

Did the author know any of these athletes prior to data collection?

2.2 Procedure and data collection

I don’t know what ‘conducting a literature review ‘means in terms of data collection. Please provide a reference justifying this or re-word.

L120-123: More information in needed on the meetings with the coaches and sport psychologists in terms of how they agreed on the interview questions. Was this guided by the researcher face to face in real time or did the participants arrange themselves?

From my understanding of the method, there were 2 sets of ‘ expert meetings ‘ .Were the same people who decided on the interview questions the same people who developed the survey questions from the interview answers?

There is no details on how the surveys were distributed and returned to the researchers?. Please provide details.

Details needed on the equipment used to record the interviews. How far apart, in terms of dates were the interviews conducted, did the same individual (researcher? ) conduct each interview?

How was the interview data, ‘used ‘ in the survey meetings by the experts. What approach was taken (provide a reference).

You say the survey (n=81) was done by contacting several teams and individuals. How was this done (e.g. email, letter, phone) and what was the total N contacts made, both to teams and individuals, sent compared to those who agreed and participated?

Were the interview participants and survey participants independent?

Where the teams contacted for the survey related to the individuals who were interviewed?

2.3 Data analysis

L135-136: Delete ..” one of the qualitative research methods.

L1401-41: This sentence is unfinished: “The collected data.”:

There needs to be more information on data analysis (e.g. what approach was taken – was it thematic using codes, sub-themes and themes, provide a reference to contextualize your data analysis approach)

RESULTS

The tables and text complement each other well and are nicely presented.

DISCUSSION

You need to reference some the existing data on burnout (from traditional sports) and compare similarities and differences with e sports.

Comments on the Quality of English Language

Dear Authors

Please observe correct sentence structure at all times. I have given some suggestions. A few of your sentences are either not complete or unclear in meaning.

Reviewer 3 Report

Comments and Suggestions for Authors

1) In the table No.2, under the category "Other" is classified "Lumber pain recurrence", which I think should be classified under the subcategory "Physical exhaustion".

2) From the results presented in the "In-depth Interview" section, only 'performance', 'overtraining', and 'interpersonal relationship' concepts are discussed. Later, in the Open-ended Survey section, the concepts "Physical and psychological Exhaustion and Career and Motivation" appear. There is a lack of explanation, did these concepts arise during an In-depth Interview or an Open-ended Survey?

Reviewer 4 Report

Comments and Suggestions for Authors

I want to express my gratitude for the opportunity to be among the first to read your paper titled "An Exploratory Study on the Conceptualization of Burnout among Professional Esports Athletes."

I am confident you can produce an inspiring and informative work if you consider my feedback.

Given the focus of the "Healthcare" journal, I believe it is necessary to provide background information on esports, including explanations of terms such as LoL, sent-down, and arena and the expansion of abbreviations.

In the methodology section, it is essential to detail the use of in-depth interviews, open survey tools (question, guide), and the data collection method.

In the discussion, it would be beneficial to indicate relevant scholarly references for the assertions made and interpret the results accordingly. It is crucial to differentiate between cultural differences arising from the sport itself and the sampling of respondents, such as variations in training camp methods and the role of military service.

Regarding the grammar, I do not understand the quotation marks at the end of the introduction. Additionally, the personal remark "I worked" in the third paragraph of the discussion seems out of place.

Reviewer 5 Report

Comments and Suggestions for Authors

The authors attempting to explore the issue related to burnout amongst the professional esport athletes based on the sample from Korea. However, there are several major issues undermined the quality of the manuscript:

1) The conceptualization of burnout on esport is still weak, the authors need to further engage with the literature related to burnout. The current discussion in the introduction is only focus on the local context in Korea. The underlining research question is also not clear.

2) There is ethical concern about disclosing too much details of the respondents (p. 3), it will be easy for the reader to identify the identity of the participants.

3) The data analysis method is not clear and need more justifications.

4) In the results section, each paragraph is just consisting of the direct quotation and without any elaboration or analysis from the authors. This style is far from satisfactory.

5) The discussion of the results was also too descriptive and lack of linking with the research question and the relevancy with the burnout.

6) The authors need to further spell out the novelty and new insight of this study.

Hence, in view of the above problems, I do not think the current form of manuscript is suitable for publication.

Comments on the Quality of English Language

Need to improve the style when reporting the qualitative data.

Round 2

Reviewer 2 Report

Comments and Suggestions for Authors

An Exploratory study on the Conceptualization of Burnout among the Professional Esports Athletes

Dear Authors

I enjoyed reading your manuscript. It is certainly improved from the 1st submission. Please find my comments below.

ABSTRACT

L18: By another author

INTRODUCTION

L57: an explanatory research….

L86-89: This sentence needs rewording – possibly split into 2 sentences.

L64: Consists of 10 teams…..

L67: Delete too

L70: I’m unclear what you mean by ‘ mental sports ‘ , do you mean psychologically demanding sports. Please insert.

L92: The perception of esports psychological demands…..

METHOD

2.1 Participants

L122: Table 1 – times is a whole number . If this unit is hours (of the interview?) please state in the table headings.

L146-143: I still don’t have a sense of the open-ended survey items. Can you give examples of the survey questions? Where they open or closed questions or a mixture of both?

Did any of the interview participants also among the 81 questionnaire participants? If so this is another possible limitation that should be mentioned in the discussion.

L152: used by the researchers for inductive categorization during meetings.

It’s a limitation that the sample of both interviews and questionnaires was completed with both current and retired players. Can you give a proportion of current versus retired players in both the interviews and questionnaires. This needs to be referred to in the discussion as a limitation.

RESULTS

The tables and text complement each other well and are nicely presented.

L189-202 & L211-228 & L237-255: The participant quotes are interesting but quite long. Not sure how this is impinging your overall word count, but perhaps these quotes could be shortened to 2-3 sentences maximum without losing the point?

The open-ended survey results are well presented.

What does 294 in the top left corner of Table 2 mean?

DISCUSSION

The discussion is quite good in terms of analysing the interview and survey results.

You need to reference more of existing research on burnout (from traditional sports) and compare similarities and differences with e sports. You have mentioned two – 27 & 28. It would be useful if you reference some more?

You need to include a paragraph, near the end of the discussion, on limitations of your study. I’ve suggested a few limitations (sampling of participants, current versus retired players and timing of the data collection in relation to the stage of the current season.

Reviewer 4 Report

Comments and Suggestions for Authors

Dear Authors,

The distinction between the national context and the sports effect is still missing, but I think it's ok now. 

Reviewer 5 Report

Comments and Suggestions for Authors

Thanks for submitting the revised manuscript. However, there are still many major issues:

1) The authors added the McGrath's four-stage stress model, to what extend it is related to the identified theme in the results section? The conceptualization of this study is still very weak.

3) The data analysis method is still not clear. The procedure needs to be justified by the existing literature. The reason for using both in-depth interview and open-ended survey is not appealing.

4) There are too many direct quotations from the respondents in the result section.

5 and 6) The discussion section still failed to address the research gap and the theorical framework that highlighted in the introduction.
